# Crowdsourced Doppler Measurements of Time Standard Stations Demonstrating Ionospheric Variability

Kristina Collins[1], John Gibbons[1], Nathaniel Frissell[2,4], Aidan Montare[3], David Kazdan[1],
Darren Kalmbach[4], David Swartz[4], Robert Benedict[4], Veronica Romanek[2], Rachel Boedicker[1],
William Liles[4,6], William Engelke[5], David G. McGaw[7], James Farmer[4], Gary Mikitin[4], Joseph Hobart[4],
George Kavanagh[4], and Shibaji Chakraborty[8]

[1]Case Western Reserve University, Glennan Bldg 9A, 10900 Euclid Avenue, Cleveland Ohio 44106
[2]Department of Physics and Engineering, University of Scranton, Scranton PA
[3]National Institute of Standards and Technology, Boulder CO
[4]HamSCI Community
[5]University of Alabama, Tuscaloosa AL
[6]Liles Innovations LLC, Reston VA
[7]Department of Physics and Astronomy, Dartmouth College, Hanover NH
[8]Center for Space Science and Engineering Research, Virginia Tech, Blacksburg VA

**Correspondence:** Kristina Collins (kd8oxt@case.edu)

**Abstract.** Ionospheric variability produces measurable effects in Doppler shift of HF (high frequency, 3-30 MHz) skywave signals. These effects are straightforward to measure with low-cost equipment and are conducive to citizen science campaigns. The Low-Cost Personal Space Weather Station (PSWS) network is a modular network of community-maintained, open-source receivers, which measure Doppler shift in the precise carrier signals of time standard stations. The primary goal of this paper is to explain the types of measurements this instrument can make and some of its use cases, demonstrating its role as the building block for a large-scale ionospheric and HF propagation measurement network which complements existing professional networks. Here, data from the PSWS network are presented for a period of time spanning late 2019 to early 2022. Software tools for the visualization and analysis of this living dataset are also discussed and provided. These tools are robust to data interruptions and to the addition, removal or modification of stations, allowing both short- and long-term visualization at higher density and faster cadence than other methods. These data may be used to supplement observations made with other geospace instruments in event-based analyses, *e.g.*, traveling ionospheric disturbances and solar flares, and to assess the accuracy of the bottomside estimates of ionospheric models by comparing the oblique paths obtained by ionospheric ray-tracers with those obtained by these receivers. The data are archived at www.doi.org/10.5281/zenodo.6622111 (Collins, 2022).

## 1 Introduction

HF (high frequency, 3-30 MHz) Doppler sounding is an established means of observing the bottomside ionosphere. Its principle of operation is straightforward: a shift in signal path length effects a corresponding Doppler shift. This information may be integrated with other ionospheric measurements to examine ionospheric variability resulting from geophysical events.

The Doppler shift $f_D$ in a received signal may be expressed as the time derivative of the phase path of the radio signal. After Chum et al. (2018):

$$f_{\mathrm{D}} = -2 \cdot \frac{f}{c} \frac{\mathrm{d}}{\mathrm{d}t} \left( \int_0^{z_R} n \cdot \mathrm{d}r \right) = -2 \cdot \frac{f}{c} \int_0^{z_R} \frac{\partial n}{\partial N} \cdot \frac{\partial N}{\partial t} \cdot \mathrm{d}r, \tag{1}$$

where $c$ is the speed of light, $n$ is the real part of refractive index for electromagnetic waves, $N$ is the electron (plasma) density, and $z_R$ is the height of reflection. This methodology is well-established in the scientific literature (Breit and Tuve, 1925; Davies et al., 1962; Jacobs and Watanabe, 1966).

In recent years, enabling technologies have become prevalent which reduce the barriers to performing precise Doppler measurements. In particular, single-board computing greatly reduces the expense and difficulty of datalogging, and readily available GPS-disciplined oscillators (GPSDOs) allow for precision timing at a price point on the order of 100 USD. The price burden for this method is also reduced by the use of existing time standard stations, such as WWV, WWVH and CHU. These stations broadcast national standard time via AM signals with precisely controlled carriers, providing ideal signals of opportunity.

Accordingly, it is now tenable to create distributed systems of HF Doppler receivers which serve as a "meta-instrument" for the observation of ionospheric disturbances, either in short term campaigns such as the one recorded in Collins et al. (2022), or in long-term data collection such as in the dataset presented herein. Such systems are readily supported by citizen scientists in the amateur radio and shortwave listening communities (Collins et al., 2021; Frissell et al., 2022b).

These data are useful to geospace scientists seeking to build a more complete picture of short term events (lasting hours to days) which occurred during the recorded timeframe, such as solar flares and geomagnetic storms. Today, frontier science investigations in these fields generally rely on combining observations from multiple instrument platforms, including total electron content estimations derived from the Global Navigation Satellite System (GNSS TEC) (Vierinen et al., 2016), incoherent scatter radar (ISR) (Nicolls and Heinselman, 2007; Zhang et al., 2021), SuperDARN radar (Nishitani et al., 2019), and vertical ionosondes (Hunsucker, 1991; Scotto et al., 2012), among others. Oblique HF sounders such as the ones used in this dataset represent one of many tools for the multi-instrument observer, and can provide direct benefit to these investigations. To wit: Satellite measurements (*e.g.*, GNSS TEC) produce height-integrated measurements from the bottomside to topside of the ionosphere, whereas the PSWS measures bottomside variability. ISRs yield range-resolved measurements of plasma parameters throughout the ionosphere, but have limited geographic coverage and cannot run constantly, primarily due to high cost of both installation and operation. While SuperDARN radars are well-established and measure parameters of the bottomside ionosphere that cannot be measured by the PSWS, SuperDARN is a pulsed system and typically has at best a 1-minute cadence. Ionosondes, too, generally have slower cadence (3-15 minutes). Vertical ionosondes produce bottomside vertical profiles for a single site. Oblique ionosondes share a measurement geometry with the Grape, but sweep in frequency, whereas the Grape monitors a single frequency with essentially continuous time resolution, which allows for monitoring short-time scale ionospheric variability along a single path. A key advantage of the PSWS is its low cost, which allows for flexible and dynamic

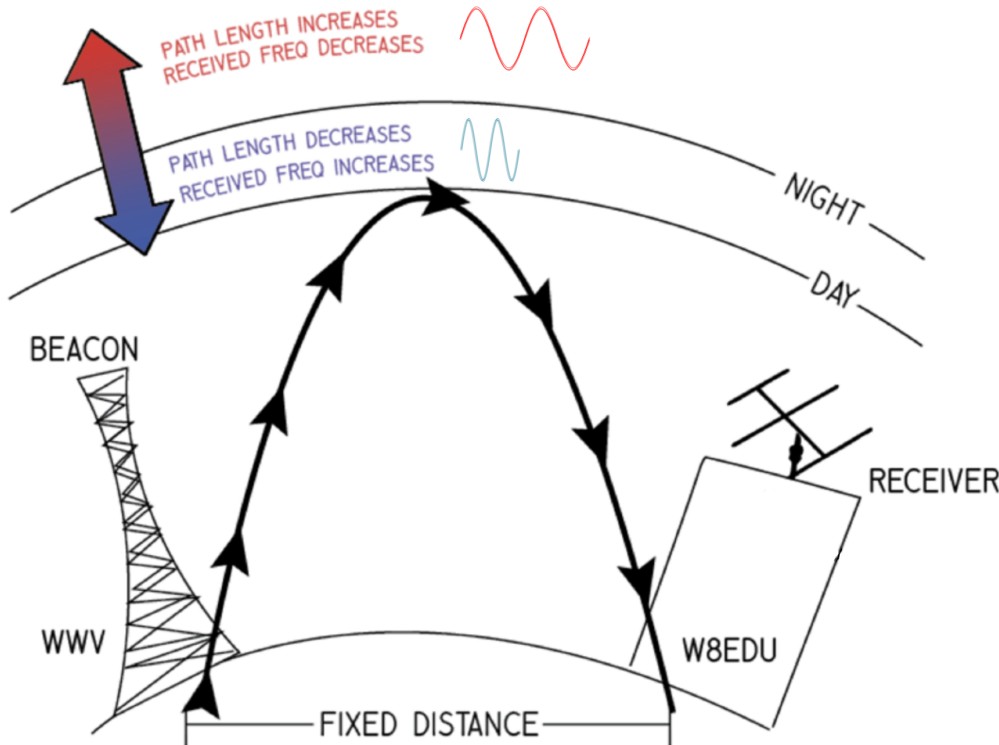

**Figure 1.** A simplified illustration of the relationship between rate of change in ionospheric layer height and received frequency shift. Precision frequency standards are required at both beacon and receiver in order to make an effective comparison. Frequency variation is generally on the order of ± 1 Hz. Multihop propagation (multiple reflections between ionosphere and ground), Pedersen modes (internal ionospheric reflections), asymmetric paths, and other factors impacting path length are not shown. Reproduced from Collins et al. (2022).

deployment of stations in regions of interest. It is also the most analogous to an HF communication system, which supports application-driven monitoring of propagation conditions.

Further insights may also be developed by examination of multiyear trends. As discussed in Section 4.3, seasonal variations are clearly evident in the longest datasets collected at the time of writing. As observations continue throughout Solar Cycle 25, we expect that these Doppler data, recorded at a greater level of coordination in the long term than has generally been achieved in the past, will support or yield novel analyses of seasonal ionospheric variability.

## 2   Background

Understanding ionospheric variability remains a frontier topic in the space physics community. This variability is key not only to understanding ionospheric dynamics in its own right, but also as a means to understanding the coupled geospace system as a whole, which includes the ionosphere's connection to both space above and the neutral atmosphere below. Ionospheric

variability takes on many forms and arises from many sources. Some forms are better understood than others. Sources of variability from space include solar flares that last minutes, *e.g.*, (Dellinger, 1937; Benson, 1964; Chakraborty et al., 2018, 2021); substorms that last a few hours, *e.g.*, (Gjerloev et al., 2007; Blagoveshchenskii, 2013; Hori et al., 2018); and ionospheric and geomagnetic storms that can last days, *e.g.*, (Buonsanto, 1999; Prölss, 2008; Thomas et al., 2016). Sources of variability from below include traveling ionospheric disturbances (TIDs) associated with atmospheric gravity waves (AGWs) *e.g.*, (Hines, 1960; Hunsucker, 1982). These are associated with terrestrial weather patterns and may be caused by events such as tornadoes (Nishioka et al., 2013), tsunamis (Galvan et al., 2011; Huba et al., 2015), or high latitude sources (Grocott et al., 2013; Frissell et al., 2016).

To understand this variability, it is important to measure over both large spatial and temporal domains and with high resolution. While many large-scale professional ionospheric sensing networks exist, the ionosphere remains significantly undersampled. To help address this undersampling issue, members of the Ham Radio Science Citizen Investigation (HamSCI) collective are working to develop the Personal Space Weather Station (PSWS), a modular, multi-instrument, ground-based space science observation platform that can be operated and afforded by individuals, as described in Collins et al. (2021, 2022). The low-cost version of the PSWS is known as the Grape, documented by Gibbons et al. (2022). The Grape is a precision, narrowband, high frequency (HF) receiver that observes ionospheric variability by measuring the Doppler shift of signals emitted by highly-stable transmitters, such as WWV and WWVH operated by the U.S. National Institute of Standards and Technology (NIST) and CHU operated by the Institute for National Measurement Standards of the National Research Council of Canada.

The Doppler shift mechanism is illustrated in Figure 1. Here, WWV transmits an HF signal that is refracted by the ionosphere back to Earth where it is received by station W8EDU. Ionospheric variability related to peak layer height, peak layer electron density, and/or layer thickness can cause changes in the propagation path that are sensed as positive Doppler shifts for decreasing path lengths (blueshifts) and negative Doppler shifts for increasing path lengths (redshifts) (Lynn, 2009). Doppler shift variations can also be used to measure the period, wavelength, and direction of TIDs (Georges, 1968; Crowley and Rodrigues, 2012; Chilcote et al., 2015; Trop et al., 2021; Trop, 2021; Romanek et al., 2022).

## 3 Methodology

### 3.1 Hardware

The majority of stations in this dataset use the purpose-built Grape V1, a low-cost receiver described by Gibbons et al. (2022). This is a low intermediate frequency receiver optimized for Doppler measurements.

It is also possible to use the software components with other hardware: as noted in the `nodelist.csv` file in the software repository, which is included in abridged form in Table A1, some of the registered nodes collect data using commercial off-the-shelf (COTS) amateur radio receivers which are capable of accepting an external frequency input from, *e.g.*, GPS-disciplined oscillators. Citizen scientists from the amateur radio and shortwave listening communities can therefore leverage their existing hardware to contribute to the PSWS network at no additional cost, and with no licensure requirement. The data processing

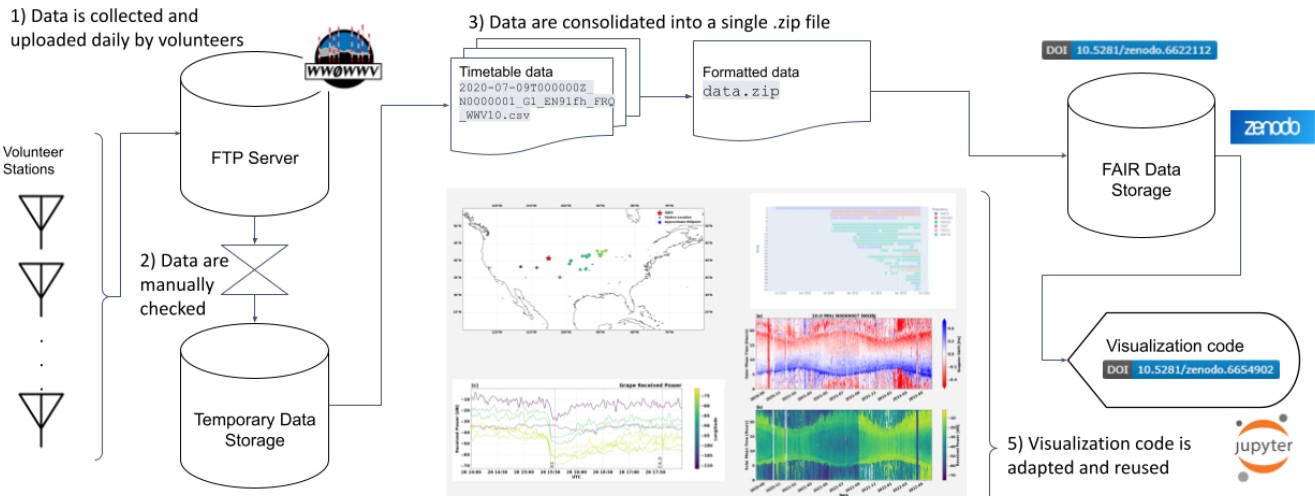

**Figure 2.** A graphical abstract showing how the data are collected, as described in Section 3.2. The visualization figures are rendered at full scale elsewhere in this paper.

framework of the Personal Space Weather Station network is robust to the addition and modification of new nodes, as well as to data outages. Data collected up to 1 June 2022 are represented in the data inventory shown in Figure 4.

### 3.2 Data acquisition process

The process of data curation is depicted in Figure 2. Each station collects 24-hour datasets according to an established standard and uploads them on a daily basis to a central FTP server. Test files, corrupted files and spurious uploads are eliminated, and the data are consolidated into a single .zip file, which is posted to the data repository (Collins, 2022). While the size of the final .zip file varies according to the number of stations collecting data, the efficiency of compression, and other factors, it is on the order of a few GB. The updated dataset can then be downloaded from Zenodo to a subdirectory in the code repository (Frissell

et al., 2022a), and used to create updated versions of the visualizations discussed in this paper.

### 3.3 File Format and Description

An example file is shown in Appendix C, which shows a file with corresponding filename `2020-07-09TT002940Z_N0000001` `_G1_EN91fh_FRQ_WWV5.csv`. This filename includes, in order: the date the data was collected and the UTC time at which that collection began; the node number, corresponding to the list in Table A1; the type of radio being used (*e.g.,* "G1" indicates

a Grape Version 1); the Maidenhead grid square in which the data was recorded; and the time standard station being measured. A detailed description of the file format and upload process is available from Collins (2021). Metadata at the beginning of each file records station information, including room for comments. The main table has three columns: UTC time, estimated frequency, and received power.

## 4 Data Visualization

The visualization code in Frissell et al. (2022a) allows for the dynamic visualization of station availability and datasets. Results can be examined on scales ranging from seconds to years, for one station in isolation or in comparison to others. Examples of this visualization code are given below. Section 4.1 describes the map and Gantt chart which summarize where and when station data is available for a given period of time. Sections 4.2 and 4.3, respectively, demonstrate short- and long-term analyses of data from a single station. Subsequent sections focus on the detection of geophysical signatures: Section 4.4 demonstrates the detection of signatures consistent with traveling ionospheric disturbances, while Section 4.5 showcases the detection of solar flares by multiple Grape stations.

### 4.1 Station Availability

A map of stations to date is shown in Figure 3. Stations were chosen on a volunteer basis, with some (*e.g.*, Node 18 in California) specifically recruited to improve coverage. Clusters of stations are evident around universities involved in the project: Case Western Reserve University in Cleveland, University of Scranton in Pennsylvania, and the New Jersey Institute of Technology each have a collection of nodes belonging to researchers. An additional cluster is generated by volunteers of the New England amateur radio community. There are also nodes close to WWV in the Fort Collins, Colorado area (*e.g.*, Node 13) which are within the transmitter's radio horizon and can be used to confirm that trends in the data originate with the ionosphere and not the radio transmitters.

Several stations are registered as nodes but do not have data included in the dataset reported at the time of writing. This may be for one of three reasons: first, the station may have data recorded but not uploaded to the FTP server; second, the station may be in the process of installing a node; third, the station may be used for experimentation with new data collection methods, including spectrum sampling and other frequency analysis algorithms. A central aspect of this work is its architecture as a living dataset, *i.e.*, a dataset into which new stations and historic data may be easily incorporated.

Figure 4 shows the data collected by each node over time. The network is modular: new stations can easily be added, and data analysis procedures are tolerant of outages and changes in frequency for each node.

### 4.2 Daily Plots

As illustrated in Figure 1, electron density in the ionosphere increases during the day as a result of photoionization and decreases at night due to recombination (Davies, 1990), producing a recognizable trend in Doppler plots. Confirming this diel variation (*i.e.*, checking for a sunrise peak) is recommended by Gibbons et al. (2022) as a benchmark for an operator to ensure that the trends observed in their station's data are geophysical in nature.

The plotting routine automatically computes the local sunrise and sunset for a given station location, and shades the background accordingly. An example of data collected by Node 1 is shown in Figure 5. The output produces two plots: Doppler shift on the top and amplitude on the bottom. In each case, the raw data, scatter-plotted in blue, undergoes filtering to produce the filtered result, which is overlaid in yellow. By default, the data processing uses a sixth-order Butterworth low-pass filter

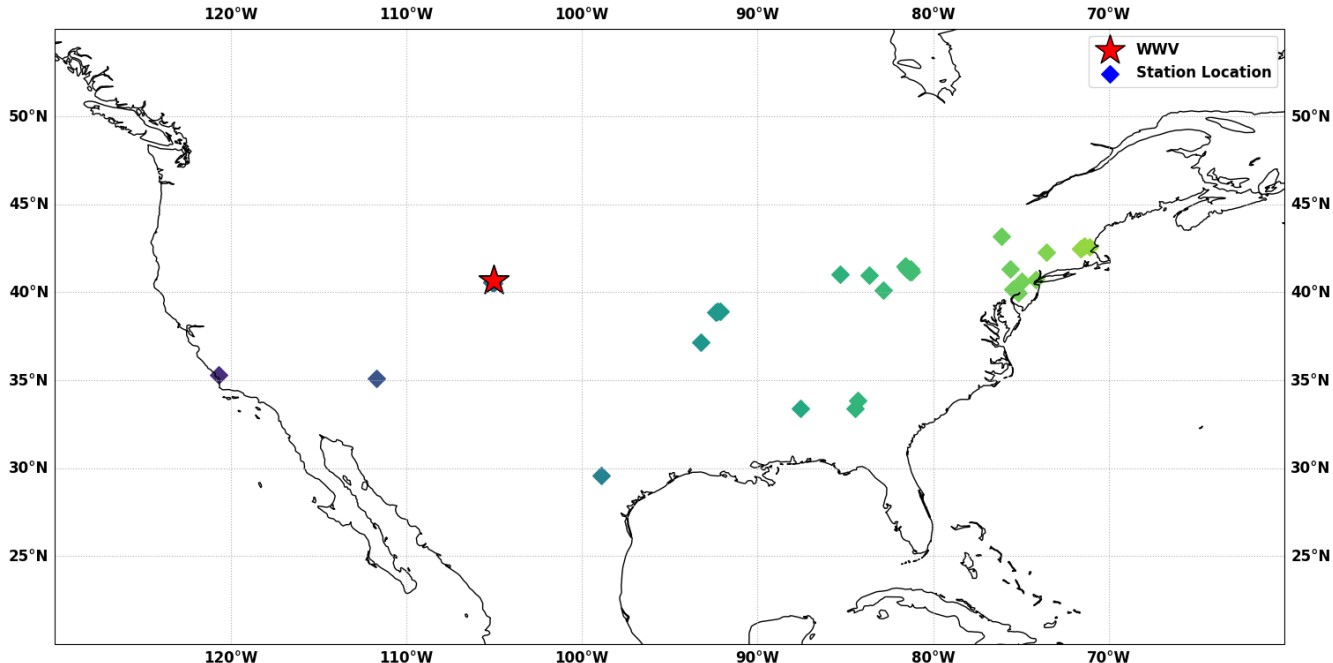

**Figure 3.** A map of currently deployed Grape stations in the United States. Per Table A1, some international stations are not shown. Scatter points mark the locations of each station. Points are color-coded by station longitude.

with cutoff frequency of 5 mHz ($T_c \approx 3.33$ minutes). A similar plotting routine is used to provide station maintainers with daily feedback, as described and depicted in Figures 10 – 13 of Gibbons et al. (2022). Diel fluctuations vary with local conditions but are distinct in long-term data, as discussed in Section 4.3. Figure 5 also shows a Doppler flash, which is discussed in Section 4.5.

### 4.3 Seasonal Climatology

Long-term trends in the data of Node 7 are shown in the time-date-parameter plots of Figure 6. Two plots are shown: Doppler shift in Hertz using a red/blue divergent colormap (*cf.* Appendix B) and received power in decibels. Each day is represented by a column of pixels, with corresponding solar mean time lined up across the plots horizontally and time's arrow running from bottom to top. On the horizontal axis, time's arrow runs left to right, covering a span from mid-2020 to spring 2022. This is consistent with the computed sun graph of Figure 7. Several observations, both geophysical and instrumentation-related in nature, can be gleaned from these two plots. First, the seasonal movement of sunrise and sunset is clearly visible at the bottom and top of the frequency plot respectively. The amplitude plot on the bottom demonstrates that reception from WWV to this station's location in the Cleveland area is much better during the nighttime, when the F2 layer of the ionosphere allows a propagation path to open up between the two locations. Vertical stripes toward the left side of both plots indicate changes or gaps in instrumentation, which are also reflected in the metadata for the affected time period. In this station's case, the station

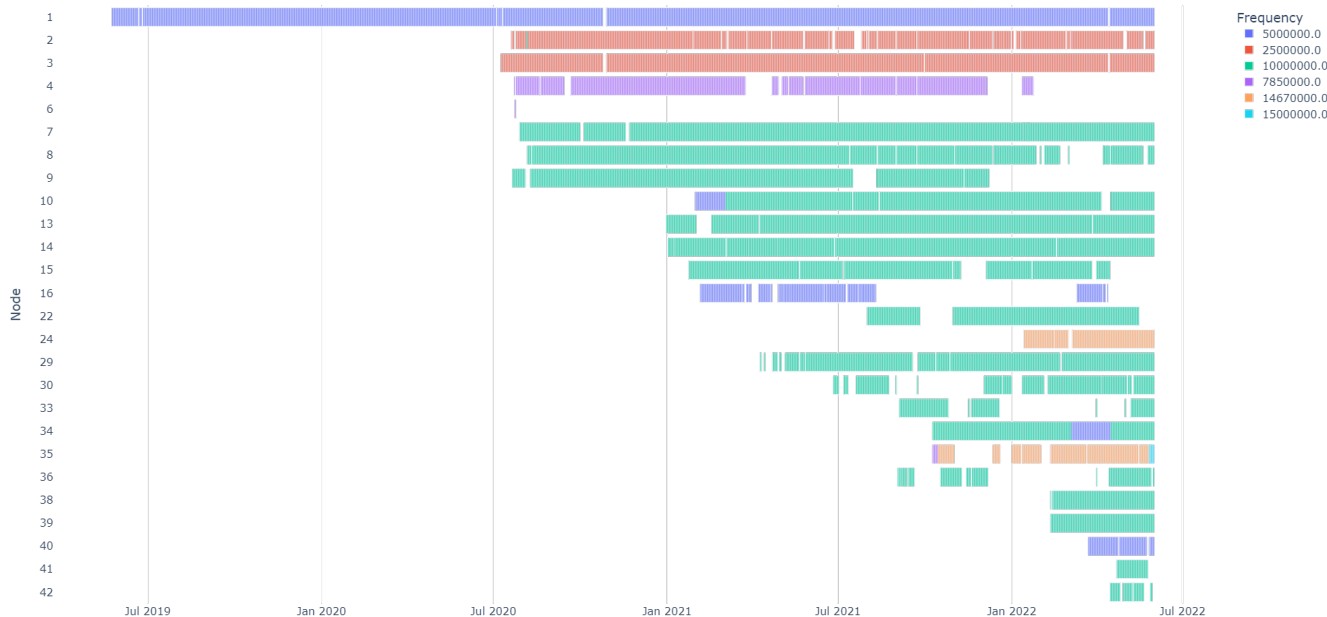

**Figure 4.** A data inventory, produced with a Gantt plotting tool in plotly express (Plotly Technologies, 2015), showing the data collected by each node.

maintainer recorded a change of antenna at his station on 26 August 2021, when he switched from an off-center-fed dipole to a magnetic loop antenna with a preamplifier. This change produced an overall increase of received power, which is clearly visible in the power plot. The lack of a corresponding change in the frequency plot above it indicates that the frequency estimation algorithm was able to function well with either antenna.

### 4.4 Traveling Ionospheric Disturbances

One category of ionospheric phenomena of particular interest for the PSWS network are medium-scale traveling ionospheric disturbances (MSTIDs), defined by Hunsucker (1982) as wavelike perturbations of ionospheric plasma with wavelengths of hundreds of kilometers, phase velocities of hundreds of meters per second and periods between 10 minutes and 1 hour. While MSTIDs may be associated with either atmospheric gravity waves (AGWs) from the neutral atmosphere (e.g., Hines, 1960; Bristow et al., 1994; Frissell et al., 2016) or from electrodynamic processes (e.g., Kelley, 2011; Atilaw et al., 2021), the source of MSTIDs is still not well understood due to their ubiquitous nature and the complexities of atmosphere-ionosphere coupling.

Trop et al. (2021) and Trop (2021) developed a technique to estimate TID period, speed, propagation direction, and velocity from a network of AM broadcast band Doppler receivers described by Chilcote et al. (2015); at the time of writing, this technique is being developed for use with HF data from the PSWS network by Romanek et al. (2022).

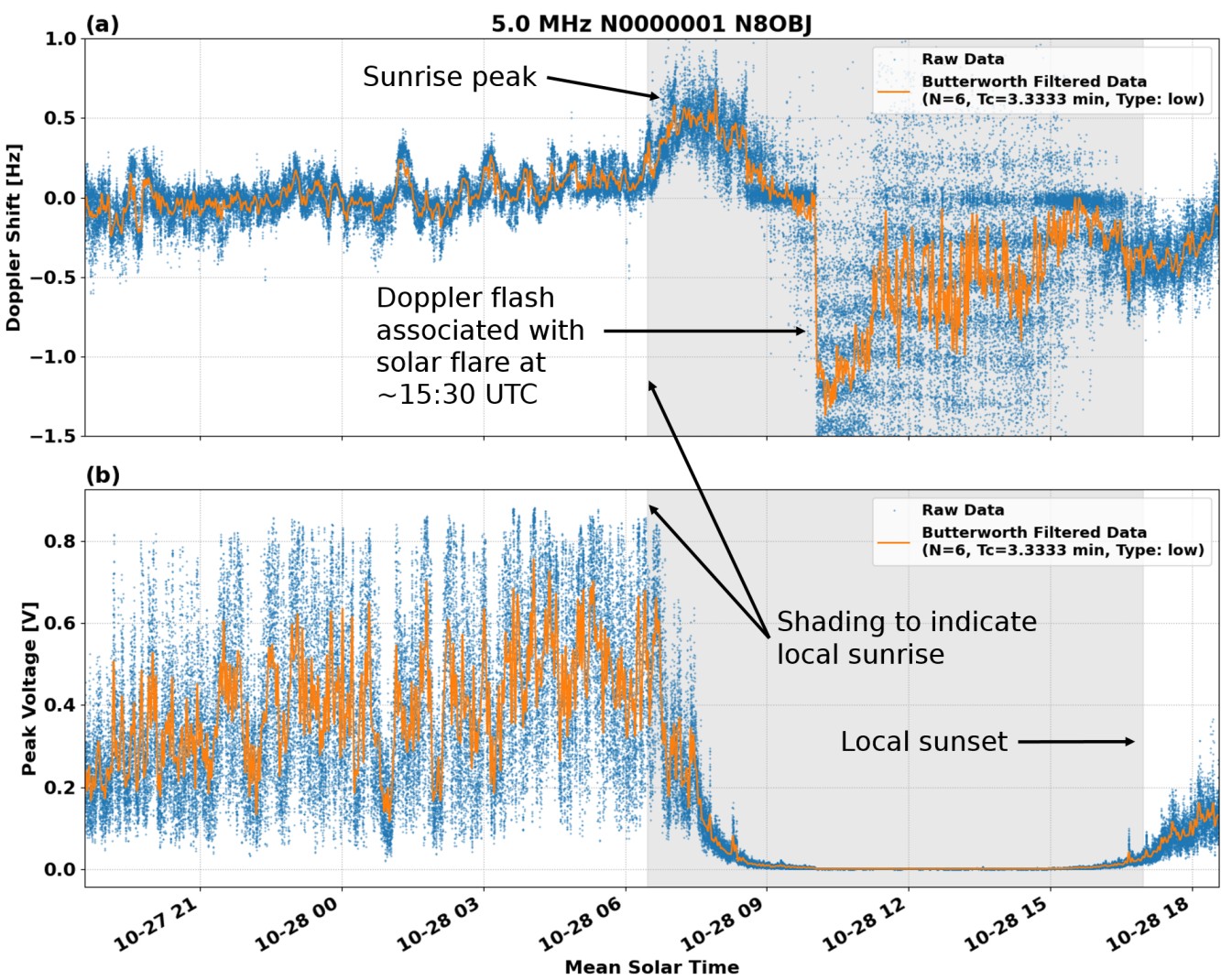

**Figure 5.** Annotated frequency and amplitude plots of Node 1's 5 MHz data from 28 October 2021, with sunrise and sunset indicated by background shading. The filtered result is superimposed on the raw data. The sunrise peak described in Gibbons et al. (2022) is clearly visible. The horizontal axis is plotted in mean solar time, rather than UTC, in order to emphasize diel effects. A Doppler flash associated with an X-class solar flare, discussed in Section 4.5, is evident around 15:30 UTC.

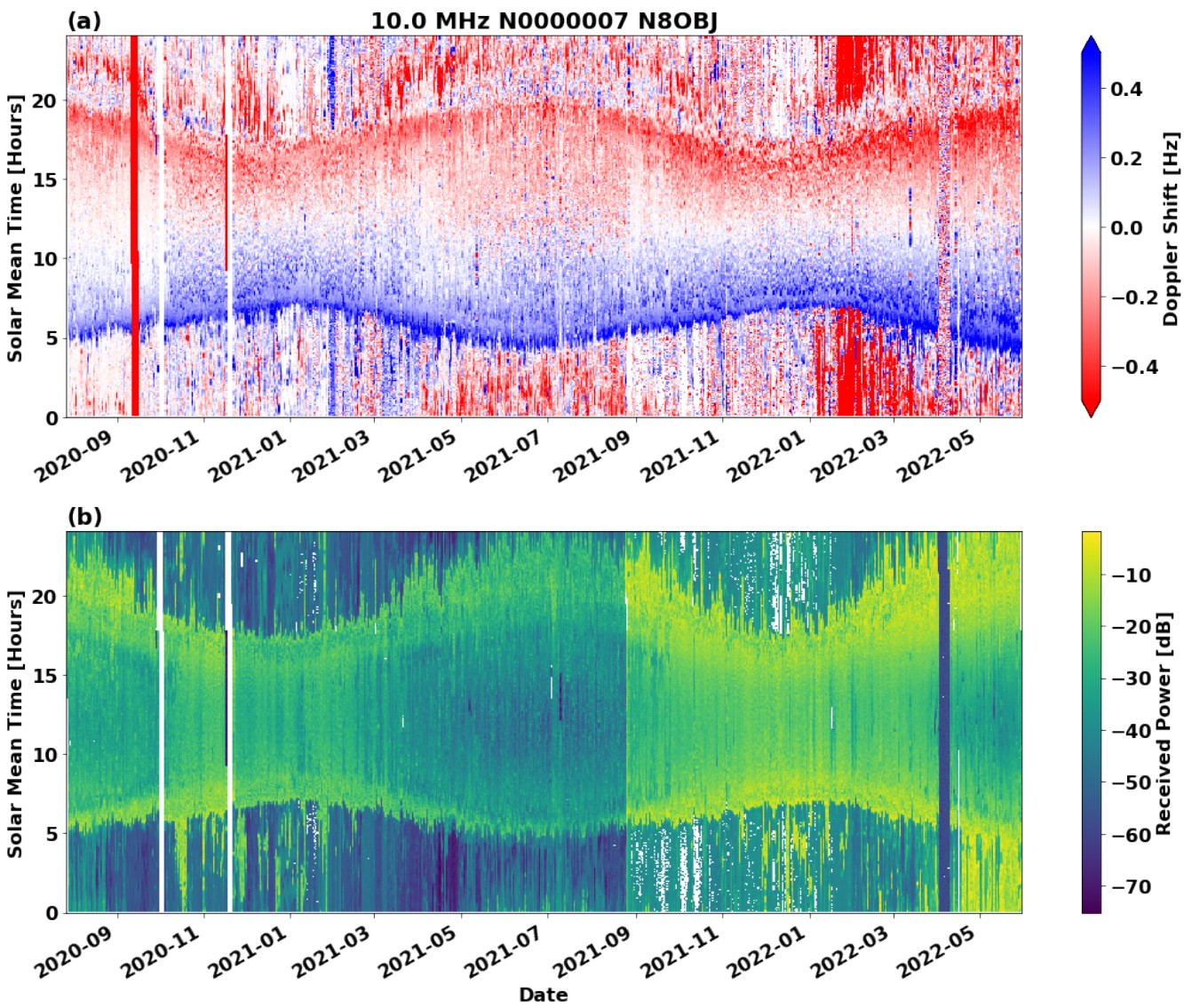

**Figure 6.** Heatmaps of frequency (top) and amplitude (bottom) at Node 7. Each day represents a line of pixels from top to bottom, with corresponding UTC times lined up across the plots horizontally. Diel variation, and the seasonal shift of sunrise and sunset times per Figure 7, is clearly visible in both plots. A new antenna and preamplifier were installed on 26 August 2021, resulting in higher received power.

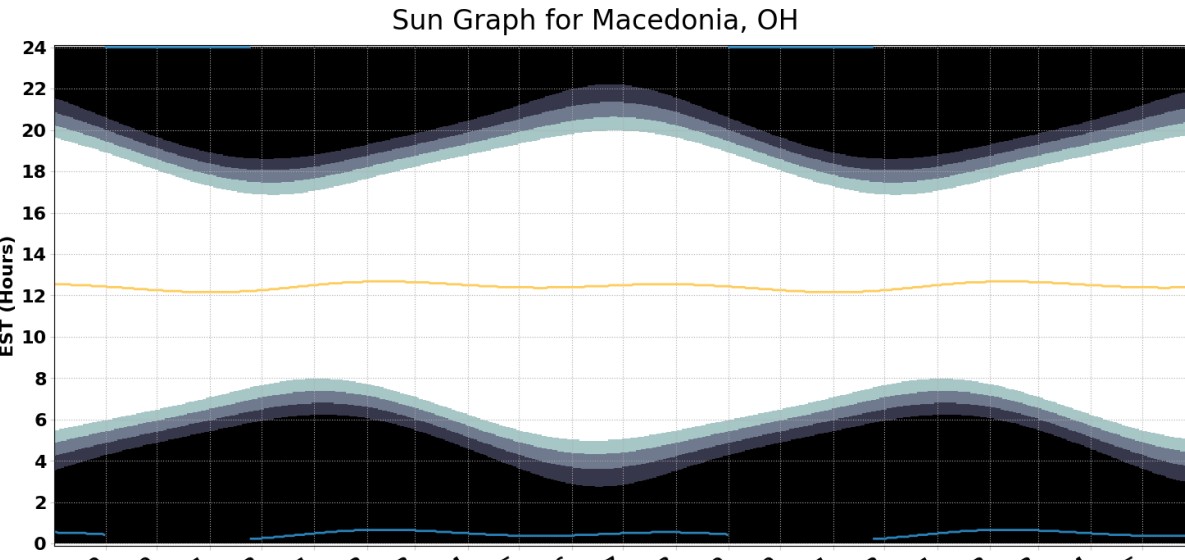

**Figure 7.** A sun graph showing sunrise and sunset times at the location of Node 7, which corresponds to the measured variation in Figure 6 (Price-Whelan, 2022).

Figure 8a is generated using the same standard processing as Figure 5. Next, the data is interpolated and filtered with a 0.5–1.2 mHz (T = 14–33 min) passband to isolate the dominant MSTID, similar to the approach used in Sect. 3.1.2 of Frissell et al. (2014). The output is then separated into four-hour bins with a ninety-percent overlap, and plotted as a spectrogram in Figure 8b, which shows signatures consistent with MSTIDs.

### 4.5 Ionospheric Response to Solar Flares

Figure 9 shows the response of the network to solar flares on 28 October 2021, providing an example of a multi-instrument measurement which demonstrates how the PSWS may augment and be validated by existing professional networks. The top plot gives the X-ray irradiance as measured by NOAA's GOES-17 spacecraft, with two flares marked: an X-class solar flare with a maximum at 15:35 UTC, and a smaller, C-class flare about two hours later. Figure 9b shows the Doppler shift for eight stations from the day in question, colormapped by station longitude; below it, Figure 9c shows the relative received power for the same stations. A longitude-dependent Doppler flash (Chakraborty et al., 2021) is observed in the frequency plot in conjunction with each flare, and a radio blackout following the X1 flare is observed in the power plot. (The lone exception, Node 13, is the groundwave station near WWV.) This Doppler flash was also measured by the SuperDARN at Fort Hays, KS, as shown in Figure 10, albeit at a slower cadence than the Grape measurements.

By default, no scaling is applied in the received power plot of Figure 9c. As discussed in Section 5.1, received signal strength varies with the antenna but may not impact the accuracy of the estimated frequency. Additionally, the PSWS nodes which use

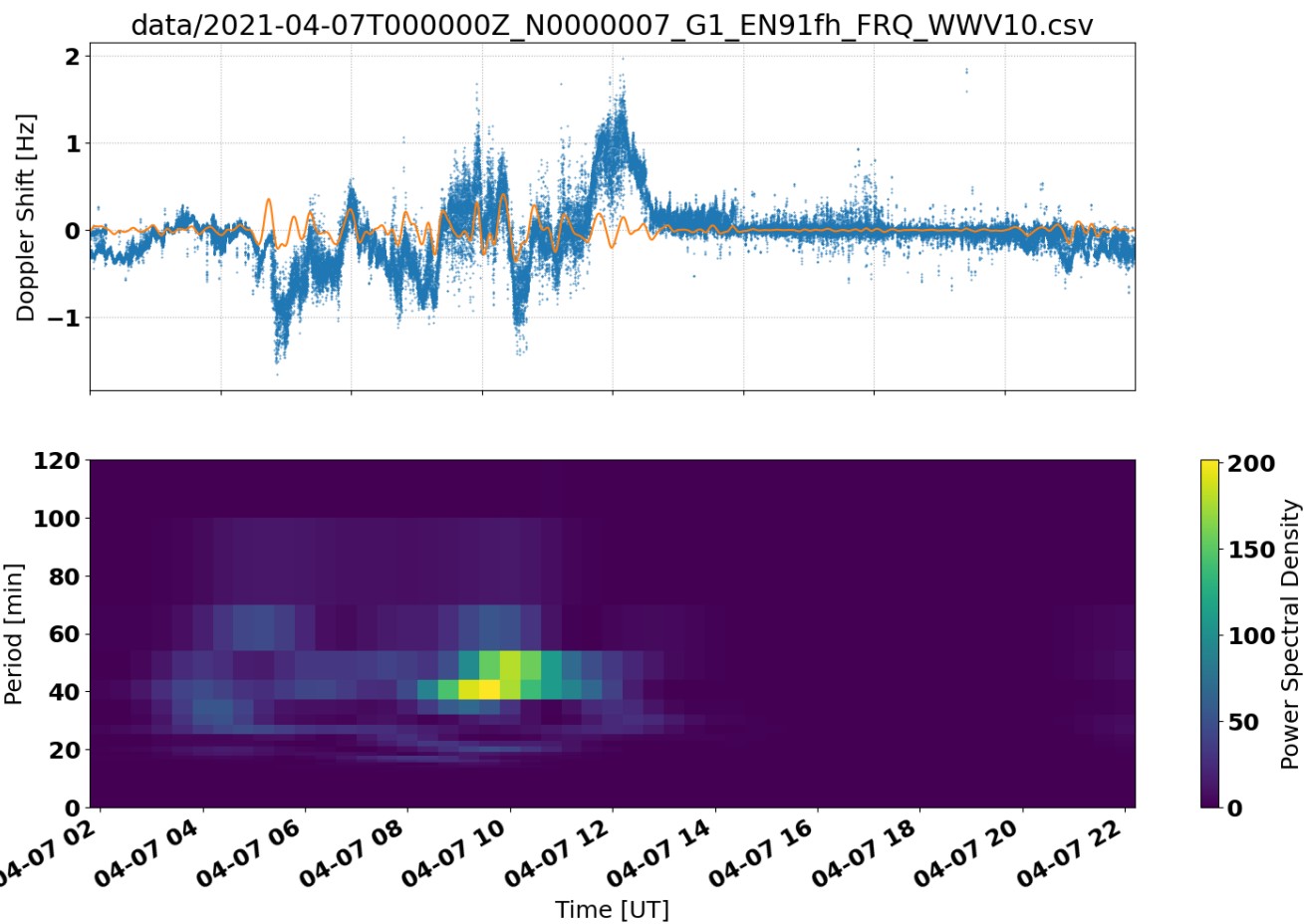

**Figure 8.** Observations of the 10 MHz WWV signal (Ft. Collins, CO) received by a Grape receiver located near Cleveland, OH on 7 April 2021 from 02-22 UT. (Top) Time series of received 10 MHz Doppler shifts. Blue dots show raw observations; orange trace shows data filtered with a 15-60 min digital bandpass Butterworth filter. (Bottom) Spectrogram showing power spectral density of the filtered data from the top panel. The oscillations and enhanced PSD in the 15-60 min band observed between ∼0330 to ∼1200 UT is consistent with signatures of medium scale traveling ionospheric disturbances.

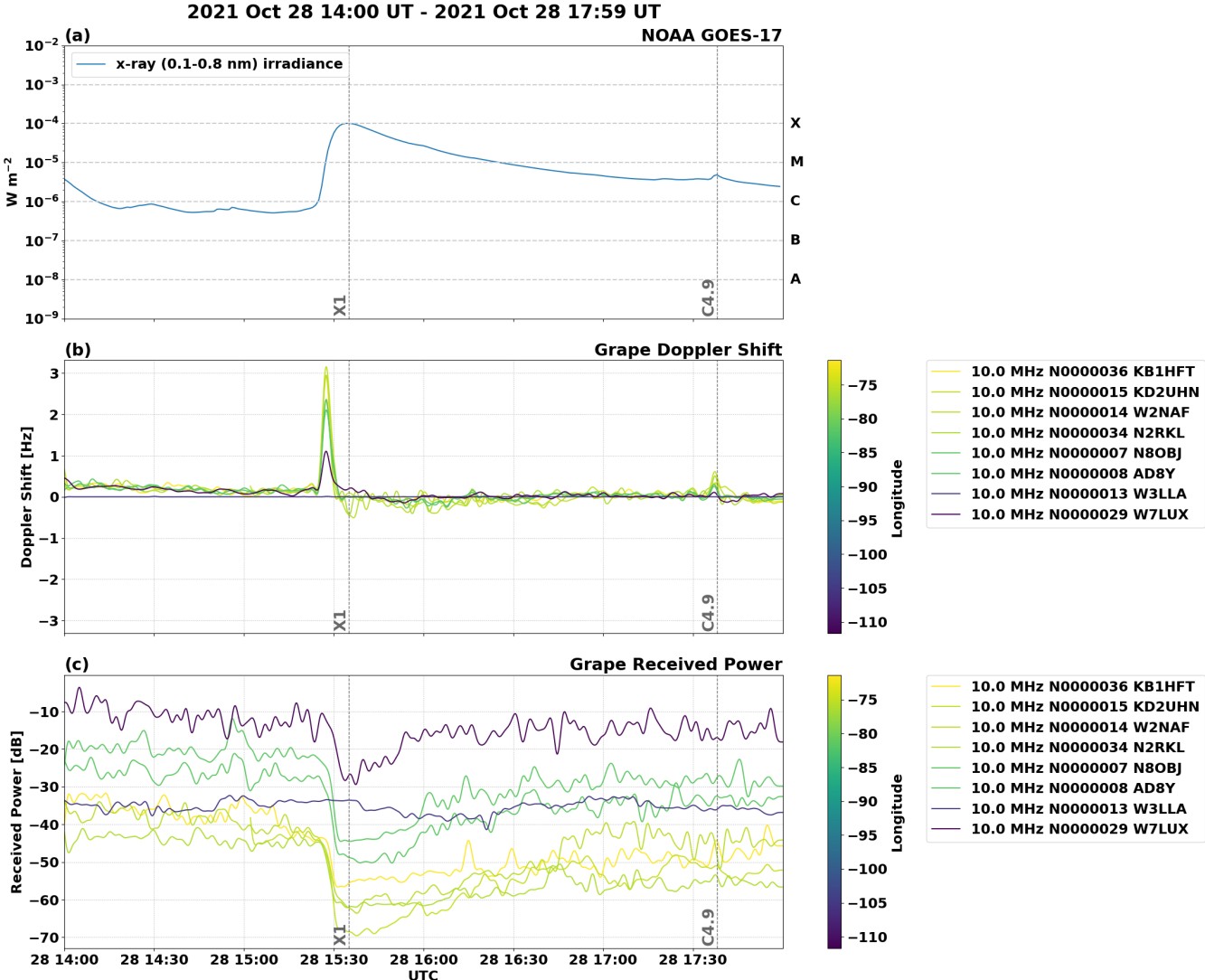

**Figure 9.** Annotated frequency and amplitude plots showing the response of Grape stations on 10 MHz to an X1 solar flare on 28 October 2021. The single-node measurement of this event on 5 MHz in Figure 5 is corroborated by other nodes in the Grape network, as well as by the SuperDARN measurement in Figure 10.

commercial off-the-shelf (COTS) hardware rather than Grape receivers (*cf.* Table A1) may have an automatic gain control (AGC) which impacts the utility of the power measurement. Therefore, users are encouraged to begin by examining the raw data from an event of interest before applying scaling.

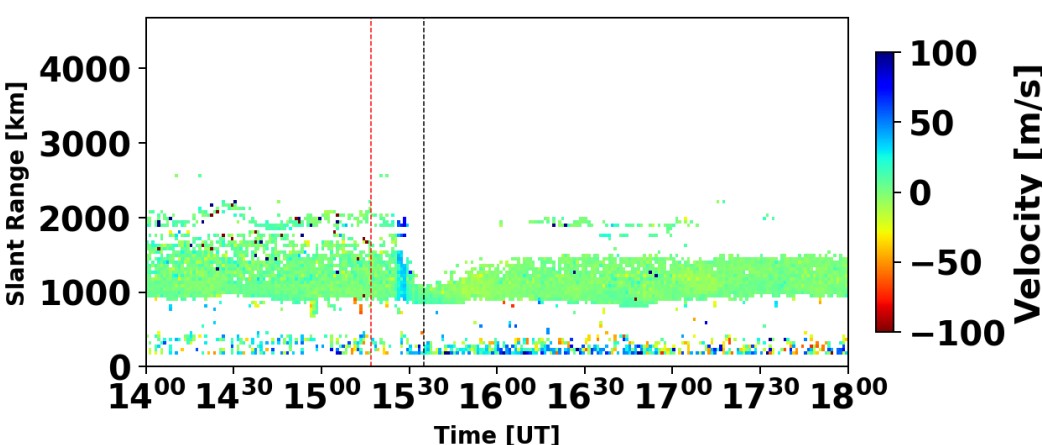

**Figure 10.** SuperDARN observation of the 28 October 2021 Doppler flash by Fort Hays East radar, beam 11. The vertical red and black lines are the start and peak of the flare, respectively. The sudden rise of velocity at ∼15:27 UT is the Doppler flash measured in Figures 5 and 9.

## 5  Discussion

### 5.1  Sources of Uncertainty

WWV's transmitter is well-characterized and inherently accurate, with a measured carrier stability below one part in $10^{12}$ (Lombardi, 2023). Allan deviation analysis by Lombardi (2022) demonstrates that the Grape V1 receiver recovers frequency with an upper bound of 2 parts per billion ($2\times10^{-10}$). Further, Lombardi performs calibration of the Leo Bodnar GPSDO recommended by Gibbons et al. (2022) and demonstrates that, with a frequency stability of one part in $10^{12}$ over a one day interval, it contributes no discernible measurement uncertainty.

Between transmitter and receiver, the received power varies according to location, antenna gain, and atmospheric attenuation. For example, in Figure 6, the antenna replacement which took place at that station in August 2021 distinctly impacts the power plot but has relatively little impact on the frequency estimation. Because the frequency and power are logged together in the raw data, the end user may elect to discard or replace frequencies where the logged power is below a threshold of their choosing. Even a low amplitude signal may yield viable frequency estimation data, however, as shown in Figure 5.

Trends observed by the network may therefore reasonably be considered to be of geophysical origin, albeit the result of multiple causes. Quantifying these ionospheric propagation effects is beyond the scope of this paper. Instead, this paper will allow investigations of these effects in the future through comparisons with other instruments and data-model comparisons.

## 5.2 Validation

We have provided comparison to prior products (*e.g.,* Breit and Tuve, 1925; Davies et al., 1962; Jacobs and Watanabe, 1966; Collins et al., 2022); event-based validation from alternate sources (GOES-17 and SuperDARN, per Figures 9 and 10 respectively), comparison of measured diel variation to model outcome (Figures 6 and 7) and initial records of sensors (*cf.* Figures 5, 9).

## 5.3 Citizen Science

The definition of citizen science has evolved over time, and the PSWS network, which invites significant personal investment and involvement from network participants, hews more closely to the "co-created" models of intensive citizen science projects, in which participants play an active role in shaping all levels of the work, than it does to "contributory" models which emphasize crowdsourced data collection (Wiggins and Crowston, 2011).

The Personal Space Weather Station exemplifies the key elements of citizen science projects identified by Pandya and Dibner (2018). They emphasize that participants in citizen science projects are primarily not project-relevant scientists; indeed, the majority of nodes in Table A1 are maintained by volunteers with no financial or academic connection to the project. Pandya and Dibner also note that citizen science projects actively engage participants, engage those participants with data, and enable those participants to derive benefit from their participation: these aspects of the PSWS are attested to by Benedict and Waugh (2021). Finally, they note that citizen science projects use a systematic approach to producing reliable knowledge, help advance science, and communicate results, all aspects which are supported by this paper.

The indispensable participation of the amateur radio and shortwave listening communities in these efforts is part of a citizen science legacy in those communities which dates back to the dawn of radio (Yeang, 2013) and continues to the present day (Collins et al., 2021).

## 6   Conclusions

1. **We present a living dataset of HF Doppler measurements made by citizen scientists.** These measurements are conducted using time standard stations' carrier signals as precise HF beacons. The amplitude and estimated Doppler shift are recorded at approximately a 1 second cadence by each station. Outages and nonstandard start times are automatically handled within the file format.

2. **A modular framework is presented for the visualization and analysis of these data.** Per Section 8, the code used to prepare the figures in this manuscript is made available for the reader's use. This code may be used to visualize future versions of the dataset as well. Additional nodes may be added to the primary dataset by coordination with the authors.

3. **Doppler data reveal both short-term and multiyear trends in ionospheric variability.** Exemplars include Figures 6 and 9 above. These data may be used in conjunction with other measurements to address frontier questions in geospace science using a multi-instrument approach.

## 7 Future Work

To date, the PSWS network comprises a growing, self-sustaining community of station maintainers. The authors foresee two means of extending this network in the future, both of which have been instrumental in fostering it to date: first, the grassroots adoption of the system by self-motivated participants, generally through amateur radio clubs; second, the targeted recruitment of station maintainers in regions of interest, particularly ahead of upcoming solar eclipses.

At the time of writing, the majority of stations are in the continental United States, but there is no inherent limitation of the system that dictates its range. The network is not limited only to Grape V1 hardware, nor to the exclusive use of WWV or other time standard stations as beacon signals. The flexible metadata format described above allows for independent signals on the amateur radio bands to be used in participatory campaigns, and for these data to be integrated seamlessly into future versions of this dataset.

Efforts are also underway to develop multichannel versions of the Grape hardware, as well as wider spectral recording to support the analysis of multiple carrier signals associated with multiple simultaneous propagation paths.

By making these data permanently accessible to professional and citizen scientists, and by continuing data collection with a growing network of stations through Cycle 25 and beyond (MacDonald et al., 2022), we hope to produce a record of short-term events and seasonal variability which will inform future studies of solar flare responses, MSTIDs and other phenomena, and which will form a benchmark for the validation of simulated Doppler shift in ionospheric models.

## 8 Code and data availability

The figures in this paper were produced using python code and Jupyter notebooks available at `https://github.com/HamSCI/hamsci_psws` (Frissell et al., 2022a). The data are available at `www.doi.org/10.5281/zenodo.6622111` (Collins, 2022). The Grape V1 hardware is fully documented in Gibbons et al. (2022), and the files to reproduce that hardware are available at Collins (2021).

## Appendix A: Table of Grape Stations

Registered nodes at the time of writing are listed in Table A1.

## Appendix B: Supplemental Colormaps

The red/blue colormap used for the frequency plot in Figure 6 is not colorblind-compliant. This colormap (`bwr_r` from Gao et al. (2015)) was chosen because it conceptually relates red and blue to red shift and blue shift, and because as a divergent colormap (saturation is lowest at times of minimal change in virtual layer height and highest at times of maximal change) it is well-suited to the data. Finding a divergent colormap which is colorblind-compliant, however, is extremely difficult, and it is not possible to ensure that it will universally meet the needs of all colorblind readers. Therefore, to improve accessibility for these

| Node | Callsign | Name | Grid Square | Latitude | Longitude | Elev. (m) | Radio | Antenna | System |
|---|---|---|---|---|---|---|---|---|---|
| 1 | N8OBJ | John C. Gibbons | EN91fh | 41.32 | -81.50 | 285 | Grape Gen 1 | DX Eng RF-PRO-1B Mag Loop /w N8OBJ preamp | RasPi3B+, FLDigi 4.1.13 |
| 2 | AD8Y | David Kazdan | EN91fl | 41.49 | -81.57 | 300 | ICOM IC-7610 | nan | RasPi3B+, FLDigi 4.1.13 |
| 3 | N8OBJ | John C. Gibbons | EN91fl | 41.32 | -81.50 | 285 | Grape Gen 1 | DX Eng RF-PRO-1B Mag Loop /w N8OBJ preamp | RasPi3B+, FLDigi 4.1.13 |
| 4 | AD8Y | David Kazdan | EN91fl | 41.49 | -81.57 | 300 | Grape Gen 1 | nan | RasPi3B+, FLDigi 4.1.13 |
| 5 | KE8HPA | Frankie Bonte | EN80nd | 40.13 | -82.84 | 275 | DXF1200 | nan | RasPi3B+, FLDigi 4.1.09 |
| 6 | KD8OXT | Kristina Collins | EN91fl | 41.49 | -81.57 | 300 | DXF1200 | nan | RasPi3B+, FLDigi 4.1.09 |
| 7 | N8OBJ | John C. Gibbons | EN91fh | 41.32 | -81.50 | 285 | Grape Gen 1 | DX Eng RF-PRO-1B Mag Loop /w N8OBJ preamp | RasPi3B+, FLDigi 4.1.13 |
| 8 | AD8Y | David Kazdan | EN91fl | 41.49 | -81.57 | 300 | Grape Gen 1 | nan | RasPi3B+, FLDigi 4.1.13 |
| 9 | KB3UMD | Aidan Montare | FN20ge | 40.17 | -75.49 | 75 | Grape Gen 1 | nan | RasPi4B, FLDigi 4.1.13 |
| 10 | KD8SYG | James Niemann | EN91ii | 41.35 | -81.28 | 330 | Grape Gen 1 | G5RV 80M 102' long | RasPi4B, FLDigi 4.1.13 |
| 11 | N8OBJ | John C. Gibbons | nan | 0.00 | 0.00 | 0 | Grape Gen 1 | nan | RasPi4B, FLDigi 4.1.13 |
| 12 | WA5FRF | Steve Cerwin | EL09nn | 29.57 | -98.88 | 412 | ICOM IC-7610 and R8600 | 30m, 40m, and 160m dipoles | Spectrum Lab, FLDIGI 4.1.14 |
| 13 | W3LLA | Maxwell Moran | DL70ln | 40.54 | -105.04 | 1511 | Yaesu FT-817 wLBGPSDO | Wire | RasPi4B, FLDigi 4.1.13 |
| 14 | W2NAF | Nathaniel A Frissell | FN21ei | 41.33 | -75.60 | 0 | Grape Gen 1 | nan | RasPi4B, FLDigi 4.1.13 |
| 15 | KD2UHN | Veronica Romanek | FN20mp | 40.63 | -74.98 | 136 | Grape Gen 1 | 80M OCF Dipole | RasPi4B, FLDigi 4.1.13 |
| 16 | WW0WWV | David A Swartz | DN70kn | 40.56 | -105.11 | 1546 | Grape Gen 1 | 20M Vertical | RasPi4B, FLDigi 4.1.13 |
| 17 | WA2UAR | Jay Silber | FM29jw | 39.95 | -75.17 | 3 | Grape Gen 1 | HD6-160 Screwdriver | RasPi4B, FLDigi 4.1.13 |
| 18 | W6BHZ | Ethan Yoshio Kita [Cal Poly] | CM95qh | 35.30 | -120.66 | 0 | Grape Gen 1 | nan | RasPi4B, FLDigi 4.1.13 |
| 19 | AB4EJ | Bill Engelke | EM63fj | 33.39 | -87.54 | 110 | Grape Gen 1 | Hexbeam ? 80m dipole | RasPi4B, FLDigi 4.1.13 |
| 20 | K2MFF | Gareth Perry | FN20vr | 40.74 | -74.17 | 50 | Grape Gen 1 | Inverted V | RasPi4B, FLDigi 4.1.13 |
| 21 | KV0S | Dave Larsen | EM38tv | 38.89 | -92.35 | 220 | nan | nan | nan |
| 22 | KD8CGH | Robert Benedict | EN91he | 41.19 | -81.33 | 300 | Grape Gen 1 | tuned small loop | RasPi4B, FLDigi 4.1.13 |
| 23 | KD0EAG | Dave Witten | EM38uw | 38.92 | -92.29 | 220 | $No_{Radio}1$ | $No_{Antenna}$ | RasPi-4, 5.4.51-v7l+ |
| 24 | PA0SLT | Drs. Wim Apon | JO33kg | 53.28 | 6.90 | 0 | Grape Gen 1 | nan | RasPi4B, FLDigi 4.1.13 |
| 25 | K2KGJ | Julius Madey | FN32fg | 42.26 | -73.54 | 372 | nan | nan | nan |
| 26 | KE8QEP | David A. Waugh | EN91id | 41.15 | -81.25 | 334 | Grape Gen 1 | Tuned Horz Dipole 10? Above gnd | RasPi4B, FLDigi 4.1.13 |
| 27 | W0DAS | David A Swartz | DN70kn | 40.56 | -105.10 | 1546 | Grape Gen 1 | 30M Dipole | RasPi4B, FLDigi 4.1.13 |
| 28 | N1JBJ | William P N Smith | FN42kn | 42.56 | -71.09 | 31 | Grape Gen 1 | 40M Homebrew Dipole | RasPi4B, FLDigi 4.1.13 |
| 29 | W7LUX | Joseph R Hobart | DM45dc | 35.09 | -111.69 | 2091 | nan | nan | nan |
| 30 | K4BSE | Jim Farmer | EM73sj | 33.39 | -84.47 | 240 | Grape Gen 1 Rcvr 1 | Loop | RasPi4B, FLDigi 4.1.13 |
| 31 | W1MTI | Vladimir A Goncharov | FN42fl | 42.49 | -71.58 | 100 | Grape Gen 1 Rcvr 1 | long wire | RasPi4B, FLDigi 4.1.13 |
| 32 | AD0RR | Todd Christell | EM37je | 37.18 | -93.23 | 382 | nan | 30 meter dipole | nan |
| 33 | AB1XB | Bill Blackwell | FN42el | 42.49 | -71.59 | 137 | Grape Gen 1 Rcvr 1 | 30M dipole | RasPi4B, FLDigi V4.1.13 |
| 34 | N2RKL | Bill Owens | FN13wd | 43.16 | -76.12 | 120 | Grape Gen 1 Rcvr 1 | magnetic loop | RasPi4B, FLDigi V4.1.13 |
| 35 | PA0RWT | Robert Wagenvoort | JO33lg | 53.25 | 6.95 | -2 | Flex 1500 | Active E-field probe; MiniWhip | RasPi3B+, Fldigi 4.1.13 |
| 36 | KB1HFT | George Kavanagh | FN42hp | 42.63 | -71.38 | 150 | Grape Gen 1 Rcvr 1 | 40m Inverted Vee | RasPi4B, Raspbian OS, FLDigi V4.1.13 |
| 37 | N8OBJ | John C. Gibbons | EN91fh | 41.32 | -81.50 | 285 | Grape Gen 2 | DX Eng RF-PRO-1B Mag Loop /w N8OBJ preamp | RasPi4B |
| 38 | WC0Y | Edward Hall (Ward) | EN71ia | 41.019 | -85.29 | 237 | Grape Gen 1 Rcvr 1 | EWE Wire | RasPi4B, Raspbian OS, FLDigi V4.1.13 |
| 39 | KM4YMI | Beau Bruce | EM73ut | 33.83 | -84.28 | 316 | Grape Gen 1 Rcvr 1 | tuned loop | RasPi3B+, Raspian OS, FLDigi V4.1.13 |
| 40 | AC0G | Michael James Hauan | EM38ww | 38.91 | -92.12 | 264 | OpenHPSDR | wire antenna | RasPi3B+, Raspian OS, FLDigi V4.1.13 |
| 41 | N8ET | Bill Kelsey | EN80ex | 40.99 | -83.65 | 243 | Grape Gen 1 Rcvr 1 | 30m vertical | RasPi4B, Raspbian OS, FLDigi V4.1.13 |

**Table A1.** Table of registered nodes at the time of writing.

plots, we have included explicit lines in the code for setting the colormap using matplotlib's colormap functions. We encourage the reader to review matplotlib's documentation at `https://matplotlib.org/stable/tutorials/colors/colormaps.html` to find an effective colormap for their needs, and to change the `bwr_r` colormap for another as required. A version of the Doppler heatmap using the viridis colormap is shown in Figure B1.

Additionally, the Colormoves interface described by Samsel et al. (2018) and available at `sciviscolor.org` allows for real-time construction and modification of colormaps using a drag-and-drop interface.

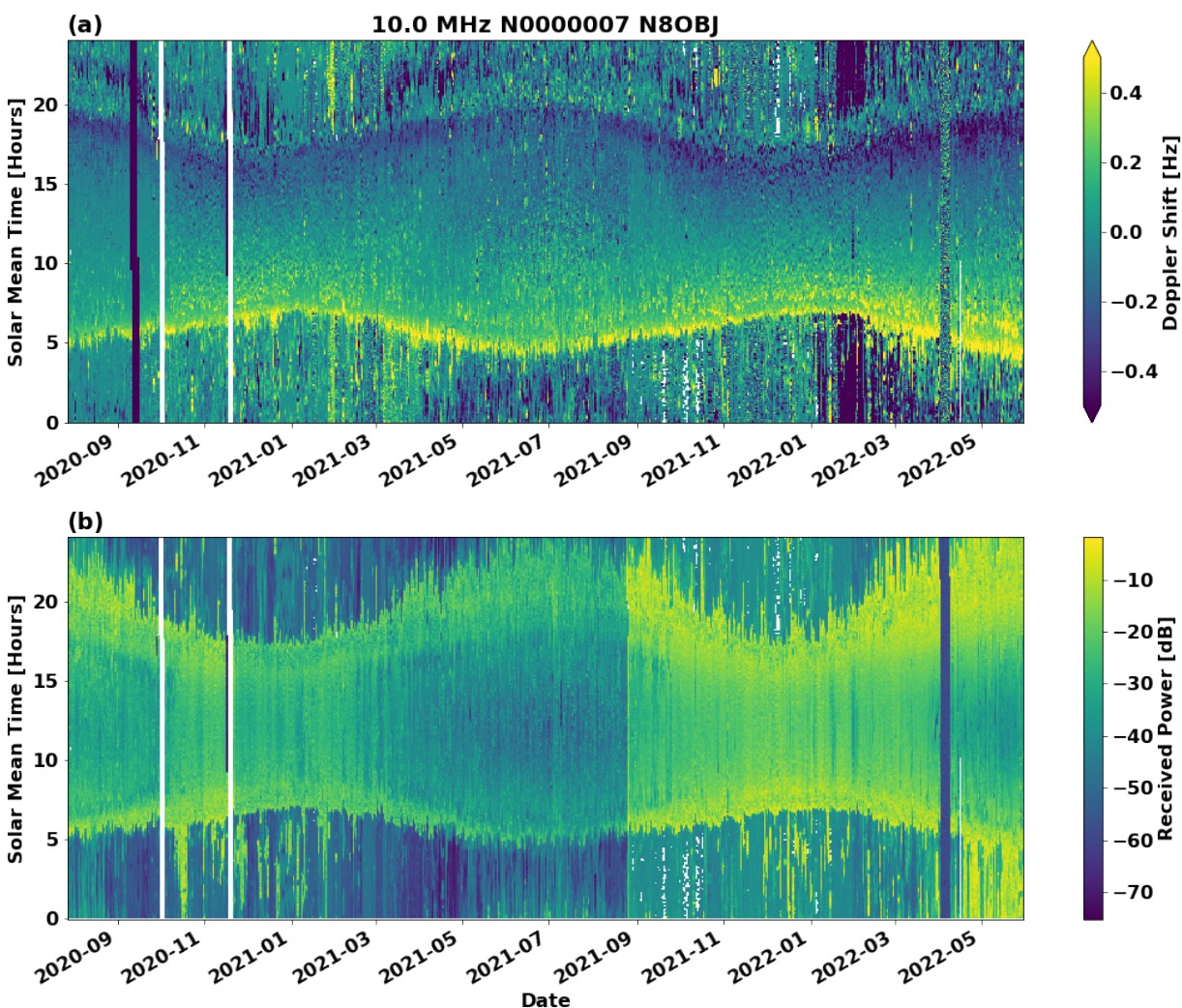

**Figure B1.** A version of Figure 6 using the perceptually uniform viridis colormap.

 **Appendix C: Example File**

The following is an example of a one-day data file with integrated metadata. This file has the filename `2020-07-09TT002940Z` `_N0000001 _G1_EN91fh_FRQ_WWV5.csv`. The file contents are self-documenting.

```
#,2020-07-09T00:29:40Z,N00001,EN91fh,41.3219273, -81.5047731, 285,Macedonia Ohio,G1,WWV5
#####################################
# MetaData for Grape Gen 1 Station
#
# Station Node Number      N00001
# Callsign                 N8OBJ
# Grid Square              EN91fh
# Lat, Long, Elv           41.3219273, -81.5047731, 285
# City State               Macedonia Ohio
# Radio1                   Grape Gen 1 Rcvr 1
# Radio1ID                 G1
# Antenna                  135 Foot OCF Dipole 30 Feet up
# Frequency Standard       LB GPSDO
# System Info              RasPi3B+, Raspian OS, FLDigi V4.1.13 (N8OBJ Modified)
#
# Beacon Now Decoded       WWV5
#
#####################################
UTC,Freq,Vpk
00:29:42,   4999999.902, 0.026468
00:29:43,   4999999.849, 0.053155
00:29:44,   4999999.838, 0.067245
00:29:45,   4999999.773, 0.065578
00:29:46,   4999999.759, 0.061869
00:29:47,   4999999.735, 0.057743
00:29:48,   4999999.800, 0.063838
00:29:49,   4999999.956, 0.088436
00:29:50,   4999999.949, 0.107922
00:29:51,   4999999.964, 0.122666
00:29:52,   4999999.956, 0.134292
```

*Author contributions.* **Kristina Collins:** Conceptualization, Software, Data Curation, Writing - Original Draft, Writing - Review & Editing, Investigation, Formal Analysis, Visualization, Project administration. **John Gibbons:** Hardware, Software, Methodology, Investigation, Resources, Project administration. **Nathaniel Frissell:** Conceptualization, Software, Visualization, Formal Analysis, Investigation, Supervision, Funding acquisition, Writing - Original Draft. **Aidan Montare:** Software, Methodology, Investigation. **David Kazdan:** Conceptualization, Methodology, Investigation. **Darren Kalmbach:** Software, Data Curation, Resources. **David Swartz:** Project administration, Investigation. **Robert Benedict:** Investigation, Software, Visualization. **Veronica Romanek:** Investigation, Formal Analysis. **Rachel Boedicker:** Investigation. **William Liles:** Investigation, Formal Analysis. **William Engelke:** Software, Writing - Revision. **David G. McGaw:** Writing - Review & Editing. **James Farmer:** Validation, Investigation, Writing - Review & Editing. **Gary Mikitin:** Validation, Investigation, Writing - Review & Editing. **Joe Hobart:** Validation, Investigation, Writing - Review & Editing. **George Kavanagh:** Investigation, Writing - Review & Editing. **Shibaji Chakraborty:** Investigation, Writing - Review & Editing, Vizualization.

*Nota bene:* This work was undertaken through the Amateur Radio Science Citizen Investigation (www.hamsci.org). Where possible, amateur radio callsigns are used herein, in addition to names, in order to specify individuals and club stations. Because these callsigns are unique and persistent identifiers, they support the Findability criterion of FAIR Data principles. The authors' callsigns are KD8OXT, N8OBJ, W2NAF, KB3UMD, AD8Y, KC0ZIE, W0DAS, KD8CGH, KD2UHN, AC8XY, NQ6Z, AB4EJ, N1HAC, K4BSE, AF8A, W7LUX, KB1HFT, and KN4BMT respectively.

*Competing interests.* The authors declare that they have no conflict of interest.

*Acknowledgements.* We gratefully acknowledge the support of NSF Grants AGS-2002278, AGS-1932997, AGS-1932972, AGS-2230345 and AGS-2230346. This work was undertaken through the HamSCI community (www.hamsci.org). We thank our volunteers for giving generously of their time and expertise, particularly the maintainers of the first contingent of Grape stations for pioneering the PSWS network and the members of the WWV Amateur Radio Club WW0WWV for their support with data curation. Hardware was constructed in the Sears Undergraduate Design Laboratory and think[box] at CWRU, with support from the Case Amateur Radio Club W8EDU. Fast Light Digital Modem Application (fldigi) is maintained by David Freese W1HKJ. We thank the developers of all python packages and open source software projects used in the analysis code documented here. This work made use of Astropy:[1] a community-developed core Python package and an ecosystem of tools and resources for astronomy (Astropy Collaboration et al., 2013, 2018, 2022). The authors acknowledge the use of SuperDARN data. SuperDARN is a collection of radars funded by national scientific funding agencies of Australia, Canada, China, France, Italy, Japan, Norway, South Africa, United Kingdom and the United States of America. We thank the reviewers for evaluating and improving this manuscript. Mike Lombardi K0WWX made significant contributions during the revision phase. Finally, we thank the staff of WWV, WWVH and CHU, without whom this work would not be possible.

---

[1]http://www.astropy.org

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
