# Peer review of "Crowdsourced Doppler Measurements of Time Standard Stations Demonstrating Ionospheric Variability"

_Earth System Science Data, 2022_

## Referee Comment (RC2)

Review ESSD-2022-303, radar detection ionosphere

Assurance: This particular data product and data description might make nice publication in ESSD. Initial flaws coupled with absence of key pieces of information, but once authors fix those and make a few other improvements, I could recommend for publication.

Authors should check ESSD guidelines, at https://www.earth-syst-sci-data.net/10/2275/2018/. Please note importance of uncertainties and validation.

I agree with substance and tone of earlier review. I repeat and amplify some of those points below.

Data easy to access and read.

Note: this reviewer prefers term 'diel' to include diurnal (daylight) and nocturnal (night-time) measurements. One advantage of diel: it does not specify mid-day or midnight maximum or minimum.

The word 'uncertainty' appears nowhere in this manuscript. Have these authors, unique in vast world of geophysics, finally achieved perfect data? Doubtful. Authors show raw and filtered data (e.g. Figs 6, 8, 9) or high (temporal) resolution data (Fig 11), with, often, actual frequency response extents of digital filters, but never an error bar. To trust and use these data, e.g. with Doppler shifts of 1 Hz, readers will need to know: variations in transmit frequency and power (authors mention at lines 104, 105 but never quantify); attenuation as it might affect frequency and power of transmitted and refracted pulses; actual refraction terms e.g. dependencies on TEC, on EC gradients, other factors; antenna gain; accuracy and uncertainty of reception; certainty of GPS reference time / oscillators (high, one hopes); etc., all for a stable medium. Then add in horizontal and vertical velocity changes on time scales from minutes to seasons, from which the authors propose detection of e.g. diel or transients patterns. A complicated chain of multiplicative uncertainties exists, from source to receiver, but authors give no hint. If authors don't quantify, how can users have confidence in their data? If, even cumulatively, uncertainties remain very small (signal to noise remains very high), or - as seems likely - uncertainty varies as a function of receiving equipment, tell us so. Prove that you know and have addressed uncertainties.

Likewise for validation. Give readers/users evidence that these measurements replicate real features. Diel patterns well known from e.g. ground-based radar, balloon-based spectrometers, satellite-based column TEC measurements? Fig 11 hints at validation (e.g. because it includes satellite data and shows both power and frequency), but authors need to provide users with validation examples. These data improve on other types? Great, show/prove it. Authors say (line 166) "discussion on validation may be found in … Gibbons". No! That paper reports hardware (receiver, frequency) performance but says nothing about seasonal patterns, vertical refraction profiles, etc. Here you want to show real data derived from these hardware systems? Prove that your data reproduce, or perhaps improve on, prior or other measurements. Validate your work!

Fig 4 shows midpoints but authors never mention, much less explain? These represent supposed refraction points/regions? Given beam dispersion and gradients of refraction index, with what horizontal or vertical uncertainties? If authors consider midpoints irrelevant, leave them out. If relevant, explain them, with uncertainties.

Fig 1 comes verbatim from Gibbon et al. 2022b. And, perhaps from other previous work from this group? Settle on definitive source, use that to establish copyright, then all subsequent uses must cite original. E.g. 'reproduced from', 'adopted from', 'modified from', your choice as

appropriate. The reader doubts clean symmetrical transmission / refraction patterns as implied in Fig 1.

Data address only Grape 1 and only source signals from WWW (Ft Collins). If authors want to mention other potential sources (necessitating different receiver frequencies), they should do so in Discussion. Including, as they do now, mention of WWVH (Hawaii) and of CHU (Canada, Ottawa), in Abstract and again in Introduction, when data only come from continental USA stations, seems misleading at best.

What do authors actually see for future of this technology? Global coverage? With what spacing? Stations outside of narrow mid-latitude regions? Again, reader gains a glimpse of network (longitudinal) spacing goals (e.g at line 103) but without follow-up or confirmation. To resolve what features? How do you improve on, validate, out-compete, etc., SuperDARN, rockets, ionosondes, satellites, etc. Any forecast possibility? Explain how systematic coverage could provide better understanding of solar impacts? How a less-expensive network compliments or replaces current capabilities. In present system, roughly half of stations inoperable at any one time (e.g Fig 5); with what impact? How does down-time impact or interact with network goals? Anchor these systems in science that you want to do. Get readers / users excited about new possibilities. Who else (outside of ionosphere / space weather communities) might use these data? If you think you have potentially a good product / good solution, give some hints about who might benefit!

Finally, we need some description other than 'citizen science'. Citizen science as ESSD promotes involves passive engagement (allow installation of weather station in garden), or active non-technical observation (standing near runway counting flights and noting tail numbers). Even CoCoRaHS, the USA NWS rain hail and snow network, which involves specific training and establishes measurement guidelines, does not require soldering, flashing of microcomputers, obtaining (at some cost in some countries) call signs (e.g line 71), transferring and uploading from SD cards, switching over (expensive) radio equipment for those using said systems, etc., as necessary for these participants. I work extensively with sensors, small networks, Arduino, etc., but I probably would not take on efforts as required here. Unless I could see real social benefit (see prior paragraph). Radio enthusiasts? Advanced community space weather trackers (ACSWT)? Not really citizen science as we understand.

A few typos exist. Authors should please give careful read as they prepare revisions. Better you and now than later at proof-reading stage.

---

## Author Comment (AC1)

Response to Reviewer 1: https://doi.org/10.5194/essd-2022-303-RC1

We thank the reviewers for their time and attention. Responses are inline below.

**In this paper, the authors describe a new experimental framework, still in development, based on the crowdsourcing paradigm for the measurement of ionospheric phenomena using radio waves. The authors focus on describing the data produced by this system and not much on the physics. Nevertheless, they outline some possible scientific questions that can be addressed as this experimental framework continues evolving.**

**Before going into the specific comments, I want to point out two general issues I see with the paper's current approach.**

**First, I found it challenging to infer the paper's primary goal. It was not explicit in either the abstract or the introduction.**
We have added this sentence to the abstract: *"The primary goal of this paper is to explain the types of measurements this instrument can make and some of its use cases, demonstrating its role as the building block for a large-scale ionospheric and HF propagation measurement network which complements existing professional networks.."*

**Second, the authors might be underestimating the potential impact of this work. As I see it, this is the first step in building a system to systematically assess the accuracy of the bottom-side estimates from almost all ionospheric models. These measurements can be compared to the oblique paths obtained with each ionospheric simulation if coupled with accurate ray-tracing solvers. If the authors agree that this is a viable application, they should mention it in the paper.**
We have added this to our abstract and Future Work sections. The suggestion is greatly appreciated.

In the abstract: *"These data may be used to supplement observations made with other geospace instruments in event-based analyses, e.g., traveling ionospheric disturbances and solar flares, and to assess the accuracy of the bottom-side estimates of ionospheric models by comparing the oblique paths obtained by ionospheric ray-tracers with those obtained by these receivers."*

In Future Work (Sect. 7): *"By making these data permanently accessible to professional and citizen scientists, and by continuing data collection with a growing network of stations through Cycle 25 and beyond (MacDonald et al., 2022), we hope to produce a record of short-term events and seasonal variability which will inform future studies of solar flare responses, MSTIDs and other phenomena, and which will form a benchmark for the validation of simulated Doppler shift in ionospheric models.*"

**The following is a list of specific comments. I will use "l." to refer**

**to "line."**

**I.1: It is unclear what you mean by "atmospheric coupling." If it is being used as an umbrella term for neutral atmosphere, solar activity, particle precipitation, etc., it might be better to say "ionospheric variability."**

The phrase "due to atmospheric coupling" has been removed. The sentence now reads *"Ionospheric variability produces measurable effects in Doppler shift of HF (high frequency, 3-30 MHz) skywave signals."*

**I.12: My understanding is that Doppler shifts can be caused not only by changes in ionospheric height. I do not think you have to make this assumption, but if you want to focus on height, you should be very specific about this being an essential assumption of the paper.**

We have edited the introduction to remove this assumption. The beginning of the introduction now reads, *"HF (high frequency, 3-30 MHz) Doppler sounding is an established means of observing the bottomside ionosphere. Its principle of operation is straightforward: a shift in signal path length effects a corresponding Doppler shift. This information may be integrated with other ionospheric measurements to examine ionospheric variability resulting from geophysical events."*

**I.23: Considering that one of the main contributions of this experimental framework is the role of citizen science, you should elaborate further on what it is and its advantages and limitations for this work.**

We have added an additional section (Section 5.3) providing context for this work in the landscape of citizen science. This topic is also addressed in the feedback to Reviewer 2.

**I.32: "Long-term ionospheric trends" is a whole area of research studying time series, often covering several solar cycles. Maybe you can use "seasonal variability."**

This change has been implemented. We have also replaced "long-term" with "seasonal," "multiyear," etc. elsewhere in the paper.

**I.45: AGWs are ubiquitous and are not restricted to the mechanisms you listed.**

We have edited this sentence for clarification: *"[AGWs] are associated with terrestrial weather patterns and may be caused by events such as tornadoes (Nishioka et al., 2013), tsunamis (Galvan et al., 2011; Huba et al., 2015), or high latitude sources (Grocott et al., 2013; Frissell et al., 2016)."*

**Figure 1: Should elaborate on what multi hops and Pederson modes are.**

The caption now reads: *"A simplified illustration of the relationship between rate of change in ionospheric layer height and received frequency shift. Precision frequency standards are required at both beacon and receiver in order to make an effective comparison. Frequency variation is generally on the order of ± 1 Hz. Multihop propagation (multiple reflections between ionosphere and ground), Pedersen modes (internal ionospheric reflections), asymmetric paths, and other factors impacting path length are not shown. Reproduced from Collins et al. (2022)."*

**I.57: Consider using a simple algebraic expression to illustrate the dependency between phase, wavelength, and local ionospheric parameters.**

We have added an equation from Chum et al (2018):

The Doppler shift $f_D$ in a received signal may be expressed as the time derivative of the phase path of the radio signal. After Chum et al. (2018):

$$f_D = -2 \cdot \frac{f}{c} \frac{d}{dt} \left( \int_0^{z_R} n \cdot dr \right) = -2 \cdot \frac{f}{c} \int_0^{z_R} \frac{\partial n}{\partial N} \cdot \frac{\partial N}{\partial t} \cdot dr, \tag{1}$$

where $c$ is the speed of light, $n$ is the real part of refractive index for electromagnetic waves, $N$ is the electron (plasma) density, and $z_R$ is the height of reflection. This methodology is well-established in the scientific literature (Breit and Tuve, 1925; Davies et al., 1962; Jacobs and Watanabe, 1966).

**I.77: What does "cleaning" imply?**

No actual changes to the data are made at this point in the process. Rather, the filenames and sizes are given a cursory review which takes only a few minutes. This sentence has been rephrased for clarification: *"Test files, corrupted files and spurious uploads are eliminated, and the data are consolidated into a single .zip file, which is posted to the data repository (…)."* The graphical abstract has also been amended to use the phrase "Data are manually checked" and "Cleaned data storage" has been redesignated "Temporary data storage."

**Figure 3: The image is too big, considering the information it is displaying. I suggest making it smaller or just listing these files' information.**

The file has been included as an appendix instead.

**I.118: Instead of displaying the frequency response of the Butterworth filter, you should limit it to summarize its features. The details shown in Figures 7 and 10 are unnecessary for the explanations presented.**

This change has been implemented.

**I.124: Instead of "line of pixels from bottom to top," you might want to use "columns."**

This change has been implemented: *"Each day is represented by a column of pixels, with corresponding solar mean time lined up across the plots horizontally and time's arrow running from bottom to top."*

**I127: You should elaborate on the mechanism responsible for this seasonal movement considering this is one of the main outputs of the measurements.**

This topic is also addressed in feedback to Reviewer 2. We have added a plot of computed sunrise/sunset times in order to reinforce the connection between the trends shown in the time/date/parameter plots and the changing length of day as the seasons progress. We have also added exposition in Section 4.2: *"As illustrated in Figure 1, electron density in the ionosphere increases during the day as a result of photoionization and decreases at night due to recombination (Davies, 1990), producing a recognizable trend in Doppler plots. Confirming*

*this diel variation (i.e., checking for a sunrise peak) is recommended by Gibbons et al. (2022) as a benchmark for an operator to ensure that the trends observed in their station's data are geophysical in nature."*

**Figure 6: Give more details on the location of the sunrise peak.**
We have added an annotation in the new version of the figure. We also reduced the range of the y axis and used a smaller marker size to make the peak more evident.
To address feedback from Reviewer 2 regarding validation, we changed the date being plotted in this example from 1 October 2019 to 28 October 2021, because it shows the Doppler flash discussed elsewhere in the paper in addition to a good sunrise peak.

**Figures 8 and 10: What are the sources of variability in Doppler shift and power? Is it just experimental uncertainty, or are there other ionospheric mechanisms involved?**
This topic has also been addressed in feedback to Reviewer 2. We have added a section on measurement uncertainty. There are of course many ionospheric mechanisms at play, but quantifying the contributions of these mechanisms goes beyond the scope of this paper and is a subject for future work.

**Figure 9: The caption needs more details. Using a smaller marker size would facilitate visualization.**
More detail has been added to this caption: "Observations of the 10 MHz WWV signal (Ft. Collins, CO) received by a Grape receiver located near Cleveland, OH on 7 April 2021 from 02-22 UT. (Top) Time series of received 10 MHz Doppler shifts. Blue dots show raw observations; orange trace shows data filtered with a 15-60 min digital bandpass Butterworth filter. (Bottom) Spectrogram showing power spectral density of the filtered data from the top panel. The oscillations and enhanced PSD in the 15-60 min band observed between ~0330 to ~1200 UT is consistent with signatures of medium scale traveling ionospheric disturbances."

A smaller marker size has also been used here, as well as in the revised version of the one-day plot above it.

**l.149: There might be better choices for a time series with such a finite perturbation than a band-pass filter. Have you considered calculating the high-frequency oscillation from the difference between the original time series and its smoothed form?**
We chose the filter used here particularly for looking at MSTIDs, which are generally defined to have periods on the order of 15 to 60 minutes. The same filter has been used by the authors before for this purpose (cf. 10.1002/2014JA019870, 10.1002/2015JA022168). The raw data is archived, so end users may apply other filters as needed for their phenomena of interest.

**Figure B1: The colormap is different from Figure 9.**
We chose to include a discussion of colormap options as an appendix. Figure B1 is intended to be analogous to the time-date parameter plot (Figure 8 in original draft).

---

## Author Comment (AC2)

Response to Reviewer 2: https://doi.org/10.5194/essd-2022-303-RC2

We thank the reviewers for their time and attention. Responses are inline below.

**Review ESSD-2022-303, radar detection ionosphere**

**Assurance: This particular data product and data description might make nice publication in ESSD. Initial flaws coupled with absence of key pieces of information, but once authors fix those and make a few other improvements, I could recommend for publication.**

**Authors should check ESSD guidelines, at https://www.earth-syst-sci-data.net/10/2275/2018/. Please note importance of uncertainties and validation.**

**I agree with substance and tone of earlier review. I repeat and amplify some of those points below.**

**Data easy to access and read.**

**Note: this reviewer prefers term 'diel' to include diurnal (daylight) and nocturnal (night-time) measurements. One advantage of diel: it does not specify mid-day or midnight maximum or minimum.**
We have adopted this word into the paper at a few points.

**The word 'uncertainty' appears nowhere in this manuscript. Have these authors, unique in vast world of geophysics, finally achieved perfect data? Doubtful. Authors show raw and filtered data (e.g. Figs 6, 8, 9) or high (temporal) resolution data (Fig 11), with, often, actual frequency response extents of digital filters, but never an error bar.**
We have added a section on uncertainty (Section 5.1) to address these concerns. For convenience, we have addressed topics raised by the reviewer inline here:

**To trust and use these data, e.g. with Doppler shifts of 1 Hz, readers will need to know: variations in transmit frequency and power (authors mention at lines 104, 105 but never quantify);**
Per Section 5.1 in the revised version: *"WWV's transmitter is well-characterized and inherently accurate, with a measured carrier stability below one part in $10^{12}$ (Lombardi, 2023)."*
Per Section 4.1: *"There are also nodes close to WWV in the Fort Collins, Colorado area (e.g., Node 13) which are within the transmitter's radio horizon and can be used to confirm that trends in the data originate with the ionosphere and not the radio transmitters."*

**attenuation as it might affect frequency and power**
Per Section 5.1 in the revised version: *"Between transmitter and receiver, the received power varies according to location, antenna gain, and atmospheric attenuation. For example, in Figure*

*6, the antenna replacement which took place at that station in August 2021 distinctly impacts the power plot but has relatively little impact on the frequency estimation. Because the frequency and power are logged together in the raw data, the end user may elect to discard or replace frequencies where the logged power is below a threshold of their choosing.*
*Even a low amplitude signal may yield viable frequency estimation data, however, as shown in Figure 5."*

**of transmitted and refracted pulses;**
The methodology used in this dataset relies on continuous monitoring of a carrier. Pulses are not used.

**actual refraction terms e.g. dependencies on TEC, on EC gradients, other factors;**
This topic is also addressed in feedback to Reviewer 1: We have added Equation 1 and relevant citations to make the connection between Doppler shift and electron density more evident.
A thorough accounting of propagation factors is outside our scope, however. Per Section 5.1 in the revised version: *"Trends observed by the PSWS may therefore reasonably be considered to be of geophysical origin, albeit the result of multiple causes. Quantifying these ionospheric propagation effects is beyond the scope of this paper. Instead, this paper will allow investigations of these effects in the future through comparisons with other instruments and data-model comparisons."*

**antenna gain; accuracy and uncertainty of reception; certainty of GPS reference time / oscillators (high, one hopes); etc., all for a stable medium.**
Per Section 5.1 in the revised version: *"Allan deviation analysis by Lombardi (2022) demonstrates that the Grape V1 receiver recovers frequency with an upper bound of 2 parts per billion ($2\times10^{-10}$). Further, Lombardi performs calibration of the Leo Bodnar GPSDO recommended by Gibbons et al. (2022) and demonstrates that, with a frequency stability of one part in $10^{12}$ over a one day interval, it contributes no discernible measurement uncertainty."*

**Then add in horizontal and vertical velocity changes on time scales from minutes to seasons, from which the authors propose detection of e.g. diel or transients patterns.**
As noted above, disambiguation of the ionospheric factors producing observed trends in these data is outside the scope of this paper; rather, that is the subject of future research that this paper exists to support.

**A complicated chain of multiplicative uncertainties exists, from source to receiver, but authors give no hint. If authors don't quantify, how can users have confidence in their data? If, even cumulatively, uncertainties remain very small (signal to noise remains very high), or - as seems likely - uncertainty varies as a function of receiving equipment, tell us so. Prove that you know and have addressed uncertainties.**
As delineated above, Section 5.1 in the revised version addresses these concerns.

**Likewise for validation. Give readers/users evidence that these measurements replicate real features. Diel patterns well known from e.g. ground-based radar, balloon-based spectrometers, satellite-based column TEC measurements? Fig 11 hints at validation (e.g. because it includes satellite data and shows both power and frequency), but authors need to provide users with validation examples. These data improve on other types? Great, show/prove it.**

We have added a figure showing a SuperDARN measurement of the Doppler flash we observed with the Grape stations. SuperDARN has a slower cadence (discussed below) so the Doppler flash is not as distinct. This serves as an example of how our network can generate a useful measurement beyond what existing professional networks are able to measure, while complementing (and being validated by) those networks.

**Authors say (line 166) "discussion on validation may be found in … Gibbons". No! That paper reports hardware (receiver, frequency) performance but says nothing about seasonal patterns, vertical refraction profiles, etc. Here you want to show real data derived from these hardware systems? Prove that your data reproduce, or perhaps improve on, prior or other measurements. Validate your work!**

We have added a section on validation (Sect. 5.2) in the revised version.

Per the ESSD guidelines referenced above by the reviewer: "(E)ach *ESSD* paper should demonstrate skill and utility of the submitted data product by some form of comparison to prior products, alternate data sources, similar products at different time or space resolution, model outcomes, initial short records of recent sensors, etc."

Per Section 5.2: *"We have provided comparison to prior products (Breit and Tuve, 1925; Davies et al., 1962; Jacobs and Watanabe, 1966; Collins et al., 2022); event-based validation from alternate sources (GOES-17 and SuperDARN, per Figures 9 and 10 respectively), comparison of diel variation to model outcome (Figures 6 and 7) and initial records of sensors (cf. Figures 5, 9)."*

**Fig 4 shows midpoints but authors never mention, much less explain? These represent supposed refraction points/regions? Given beam dispersion and gradients of refraction index, with what horizontal or vertical uncertainties? If authors consider midpoints irrelevant, leave them out. If relevant, explain them, with uncertainties.**

We thank the reviewer for highlighting this. The midpoints have been removed from the map. They are a first approximation inviting a more intensive propagation discussion outside the scope of this paper, so we plan to discuss them in a future publication instead.

**Fig 1 comes verbatim from Gibbon et al. 2022b. And, perhaps from other previous work from this group? Settle on definitive source, use that to establish copyright, then all subsequent uses must cite original. E.g. 'reproduced from', 'adopted from', 'modified from', your choice as appropriate. The reader doubts clean symmetrical transmission / refraction patterns as implied in Fig 1.**

These changes have been implemented. The caption now reads: "A simplified illustration of the relationship between shift in ionospheric layer height and received frequency shift. Precision frequency standards are required at both beacon and receiver in order to make an effective comparison. Frequency variation is generally on the order of ±1 Hz. Multihop propagation (multiple reflections between ionosphere and ground), Pedersen modes (internal ionospheric reflections), and asymmetric paths are not shown. Reproduced from Collins et al. 2022b)."

**Data address only Grape 1 and only source signals from WWW (Ft Collins). If authors want to mention other potential sources (necessitating different receiver frequencies), they should do so in Discussion. Including, as they do now, mention of WWVH (Hawaii) and of CHU (Canada, Ottawa), in Abstract and again in Introduction, when data only come from continental USA stations, seems misleading at best.**
The mention of these stations has been removed from the abstract. Although the data collected do principally come from stations in the continental US, the data inventory plot shows that there are three stations (Nodes 4, 24 and 35) collecting data from CHU. Additionally, WWV and WWVH share carrier frequencies, so all three stations are relevant to the data collected here.

**What do authors actually see for future of this technology? Global coverage? With what spacing? Stations outside of narrow mid-latitude regions? Again, reader gains a glimpse of network (longitudinal) spacing goals (e.g at line 103) but without follow-up or confirmation. To resolve what features?**
This topic is also addressed in feedback to Reviewer 1. We have added a Future Work section to address these topics: *"To date, the PSWS network comprises a growing, self-sustaining community of station maintainers. The authors foresee two means of extending this network in the future, both of which have been instrumental in fostering it to date: first, the grassroots adoption of the system by self-motivated participants, generally through amateur radio clubs; second, the targeted recruitment of station maintainers in regions of interest, particularly ahead of upcoming solar eclipses.*

*At the time of writing, the majority of stations are in the continental United States, but there is no inherent limitation of the system that dictates its range. The network is not limited only to Grape V1 hardware, nor to the exclusive use of WWV or other time standard stations as beacon signals. The flexible metadata format described above allows for independent signals on the amateur radio bands to be used in participatory campaigns, and for these data to be integrated seamlessly into future versions of this dataset.*

*Efforts are also underway to develop multichannel versions of the Grape hardware, as well as wider spectral recording to support the analysis of multiple carrier signals associated with multiple simultaneous propagation paths.*

*By making these data permanently accessible to professional and citizen scientists, and by continuing data collection with a growing network of stations through Cycle 25 and beyond (MacDonald et al., 2022), we hope to produce a record of short-term events and seasonal variability which will inform future studies of solar flare responses, MSTIDs and other phenomena, and which will form a benchmark for the validation of simulated Doppler shift in ionospheric models."*

Regarding spacing: the network is intended to be largely organic, as discussed in Sections 5.3 and 7, and valid data may still be collected by a handful of stations. As yet, no straightforward density requirements are imposed by current scientific goals. (Further discussion below re: systematic coverage.)

**How do you improve on, validate, out-compete, etc., SuperDARN, rockets, ionosondes, satellites, etc.**
We have added the following to the introduction: *"Oblique HF sounders such as the ones used in this dataset represent one of many tools for the multi-instrument observer, and can provide direct benefit to these investigations. To wit: Satellite measurements (e.g., GNSS TEC) produce height-integrated measurements from the bottomside to topside of the ionosphere, whereas the PSWS measures bottomside variability. ISRs yield range-resolved measurements of plasma parameters throughout the ionosphere, but have limited geographic coverage and cannot run constantly, primarily due to high cost of both installation and operation. While SuperDARN radars are well-established and measure parameters of the bottomside ionosphere that cannot be measured by the PSWS, SuperDARN is a pulsed system and typically has at best a 1-minute cadence. Ionosondes, too, generally have lower cadence (3-15 minutes). Vertical ionosondes produce bottom-side vertical profiles for a single site. Oblique ionosondes share a measurement geometry with the Grape, but sweep in frequency, whereas the Grape monitors a single frequency with essentially continuous time resolution, which allows for monitoring short-time scale ionospheric variability along a single path. A key advantage of the PSWS is its low cost, which allows for flexible and dynamic deployment of stations in regions of interest. It is also the most analogous to an HF communication system, which supports application-driven monitoring of propagation conditions."*

The region of interest for rockets is below that of this instrument, and rocket soundings cannot provide long term time series data of, *e.g.,* diel patterns.

**Any forecast possibility?**
There is potential for nowcasting/forecasting, particularly with regard to real-time HF communications links, as noted above. We have not yet implemented this functionality for the PSWS network, and so did not claim it in this paper.

**Explain how systematic coverage could provide better understanding of solar impacts?**
The discussion of solar flare response in Section 4.5, which has been significantly extended in the revised version, is intended to lay the groundwork for future work in this area. Our ongoing investigation of Doppler flash resulting from solar flares will require us to determine what such systematic coverage would look like, particularly for further investigation of the longitudinal dependence indicated in the multi–instrument plot. However, the question of coverage requirements is a complex one outside our present scope. Moreover, it will vary according to the scientific objective at hand (e.g., MSTID detection, eclipse effects), and it is therefore best handled on a case-by-case basis in future studies using these data as the network continues to evolve.

**How a less-expensive network compliments or replaces current capabilities.**
See excerpt from revised introduction above.

**In present system, roughly half of stations inoperable at any one time (e.g Fig 5); with what impact? How does down-time impact or interact with network goals?**
Per Section 4.1: *"Several stations are registered as nodes but do not have data included in the dataset reported at the time of writing. This may be for one of three reasons: first, the station may have data recorded but not uploaded to the FTP server; second, the station may be in the process of installing a node; third, the station may be used for experimentation with new data collection methods, including spectrum sampling and other frequency analysis algorithms. A central aspect of this work is its architecture as a living dataset, i.e., a dataset into which new stations and historic data may be easily incorporated."*
Figure 5 shows the network building and gaining stations over time. As with all networks, downtime is not desirable. However, as discussed above, no straightforward density requirements are imposed by current scientific goals.

**Anchor these systems in science that you want to do.**
Two specific scientific objectives are identified in the paper: MSTIDs (Section 4.4)  and solar flare response (Section 4.5). We are hopeful that the publication of this paper will support our work on these topics, which are subjects of current research in our group.

**Get readers / users excited about new possibilities. Who else (outside of ionosphere / space weather communities) might use these data? If you think you have potentially a good product / good solution, give some hints about who might benefit!**
This topic is also addressed in feedback to Reviewer 1. We have added the following to the abstract: *"The primary goal of this paper is to explain the types of measurements this instrument can make and some of its use cases, demonstrating its role as the building block for a large-scale ionospheric and HF propagation measurement network which complements existing professional networks … These data may be used to supplement observations made with other geospace instruments in event-based analyses, e.g., traveling ionospheric disturbances and solar flares, and  to assess the accuracy of the bottom-side estimates of ionospheric models by comparing the oblique paths obtained by ionospheric ray-tracers with those obtained by these receivers."* As noted above, we have also added a section on future work, and alluded in the introduction to the possible use of the Grape for application-driven monitoring of HF propagation conditions.

**Finally, we need some description other than 'citizen science'. Citizen science as ESSD promotes involves passive engagement (allow installation of weather station in garden), or active non-technical observation (standing near runway counting flights and noting tail numbers). Even CoCoRaHS, the USA NWS rain hail and snow network, which involves specific training and establishes measurement guidelines, does not require soldering, flashing of microcomputers, obtaining (at some cost in some countries) call**

**signs (e.g line 71), transferring and uploading from SD cards, switching over (expensive) radio equipment for those using said systems, etc., as necessary for these participants. I work extensively with sensors, small networks, Arduino, etc., but I probably would not take on efforts as required here. Unless I could see real social benefit (see prior paragraph). Radio enthusiasts? Advanced community space weather trackers (ACSWT)? Not really citizen science as we understand.**

This topic was also addressed in the feedback to Reviewer 1, and is discussed in Section 5.3.

In seeking a working definition of "citizen science", we take as authoritative the National Academies' report on citizen science, which in addition to establishing those traits held in common across citizen science projects also identifies several "axes across which citizen science might vary" (cf. https://nap.nationalacademies.org/read/25183/chapter/4#36). These axes include duration of participation, which most nearly encompasses the concern that the reviewer raises here. The report affirms that dedicated efforts by enthusiasts and hobbyists such as birders and amateur astronomers are indeed encompassed by the citizen science category, stating that "citizen science provides opportunities for a range of different kinds of participants, from social individuals to those less interested in ongoing social interaction, and from individuals who sample widely to those who dive deeply into a single pursuit."

**A few typos exist. Authors should please give careful read as they prepare revisions. Better you and now than later at proof-reading stage.**

We have carefully reviewed the final draft of the paper and hope to have caught as many typos as possible. We thank the reviewers, editors and staff for their dedication in helping us share this work with the scientific community.

---

## Author Response (AR2)

Author Response - essd-2022-303, final version

Thank you for the opportunity to publish this work in *ESSD.*

We have made only one change to this manuscript from the most recent revised version: Country names have been added in the author credits.

Dr. Carlson's post included the following final reviewer comment:

**Line 92: I think you mean 'no additional license requirements'. Most radio operators will have already applied for and obtained necessary operating license.**

The original sentence reads: *"Citizen scientists from the amateur radio and shortwave listening communities can therefore leverage their existing hardware to contribute to the PSWS network at no additional cost, and with no licensure requirement."*

We did not use the word "additional" in the original manuscript, since shortwave listeners are not necessarily licensed. Either version is correct, so we leave it to the editors' discretion whether to make this change.